# Reference Intervals for Coagulation Parameters in Developmental Hemostasis from Infancy to Adolescence

**DOI:** 10.3390/diagnostics12102552

**Published:** 2022-10-20

**Authors:** Giovina Di Felice, Matteo Vidali, Gelsomina Parisi, Simona Pezzi, Alessandra Di Pede, Giulia Deidda, Matteo D’Agostini, Michaela Carletti, Stefano Ceccarelli, Ottavia Porzio

**Affiliations:** 1Clinical Laboratory Unit, Bambino Gesù Children’s Hospital, IRCCS, 00165 Rome, Italy; 2Clinical Chemistry Unit, Fondazione IRCCS Ca’ Granda Ospedale Maggiore Policlinico, 20122 Milan, Italy; 3Neonatal Intensive Unit, Bambino Gesù Children’s Hospital, IRCCS, 00165 Rome, Italy; 4Department of Experimental Medicine, University of Rome Tor Vergata, 00133 Rome, Italy

**Keywords:** pediatrics, coagulation, reference interval

## Abstract

*Background*: The objective of this study was to establish the age and sex-dependent reference intervals for coagulation assays evaluated in healthy children, ranging from 0 days to 16 years old. *Methods*: PT, aPTT, Fibrinogen (functional), Antithrombin activity, Protein C anticoagulant activity, Protein S free antigen, Thrombin time, D-Dimer, Von Willebrand Factor antigen, Lupus anticoagulant (screening), extrinsic and intrinsic pathway factors, and activated Protein C resistance were evaluated using STA-R Max^2^. *Results*: A total of 1280 subjects (671 males and 609 females) were divided into five groups, according to their age: 0–15 days (n = 280, 174 M and 106 F), 15–30 days (n = 208, 101 M and 107 F), 1–6 months (n = 369, 178 M and 191 F), 6–12 months (n = 214, 110 M and 104 F), and 1–16 years (n = 209, 108 M and 101 F). The 95% reference intervals and the 90% CI were established using the Harrell–Davis bootstrap method and the bootstrap percentile method, respectively. *Conclusions*: The present study supports the concept that adult and pediatric subjects should be evaluated using different reference intervals, at least for some coagulation tests, to avoid misdiagnosis, which can potentially lead to serious consequences for patients and their families, and ultimately the healthcare system.

## 1. Introduction

Hemostasis is a complex physiological mechanism, which includes procoagulant and anticoagulant proteins. The levels of procoagulant and anticoagulant proteins differ between infants and adults, due to the hemostatic system development over time, from neonatal to adult age, with continuous changes of biological development [1,2]. The hemostatic system is not fully mature before 3 to 6 months of age; it is therefore important to acknowledge that the observed differences between adults and infants are probably physiological and do not always underlie pathologic conditions.

The coagulation system of children evolves with age; the term “developmental hemostasis” [3] describes the evolution of the coagulation system, from fetal life to adolescence [4,5,6]. Developmental hemostasis provides a protective mechanism for neonates and children, resulting in a decreased risk of thromboembolic and/or hemorrhagic events in these age groups [7].

Accurate diagnosis of inherited bleeding and thrombotic disorders requires appropriate reference intervals for coagulation parameters. At birth, the plasmatic concentrations of many coagulation proteins are around 50% of adult levels, with lower levels in preterm newborns compared to full-term, and they reach adult values by 6 months of age [3]. Moreover, several reports indicated that the plasma concentrations of individual coagulation proteins or inhibitors may not reach adult levels until early or late childhood [8,9,10,11]. There is evidence that pediatric reference intervals differ not only from adults, but are also associated to age-related developmental changes, from fetal life through to adolescence. Several studies have shown age-related quantitative differences in the levels of hemostatic proteins; in addition, pediatric reference intervals are influenced by ethnicity, gender, analyzer, reagent, and methodology [12,13].

Accurate laboratory reference intervals are pivotal for an accurate clinical decision and have a significant impact on the overall quality of patient care; in this line, the Scientific and Standardization Committee of the International Society on Thrombosis and Haemostasis (ISTH) strongly recommends that each laboratory defines the age-dependent reference ranges under its own technical conditions [14].

In this study, we report the age and sex-dependent reference intervals for common coagulation assays from an apparently healthy population of children, ranging from 0 days to 16 years old, according to the C28-A3c guidelines [15].

## 2. Materials and Methods

### 2.1. Subjects

This study was approved by the institutional ethics committee of the Bambino Gesù Children’s Hospital (certificate no. 1078_OPBG_2016), in accordance with the tenets of the Helsinki Declaration and has been approved by the authors’ institutional review board. Written informed consent was obtained from parents and/or guardians of the children. Patients scheduled to undergo minor elective surgery in the Departments of Pediatric Surgery, Otorhinolaryngology, and Urology were considered eligible for enrollment in the study, as were subjects referred to the outpatient collection center. Neonatal samples were collected from healthy term neonates. All the healthy newborns received 1 mg of vitamin K1 by intramuscular injection or orally at birth, according to the different hospital protocols, following the recommendations of the Italian Society of Neonatology (3 × 2 mg of vitamin K1 orally at birth, at 4 to 6 days and at 4 to 6 weeks).

Inclusion criteria were as follows:birth weight of >2500 g;gestation length >37 weeks;APGAR at 5 min ≥7;

Exclusion criteria were as follows:history of bleeding or thrombotic disorders;anticoagulant treatment;acute or chronic infections within 1 month prior to enrolment;presence of a disease that could affect the coagulation system (e.g., hematological diseases, liver cirrhosis);icteric, hemolyzed or lipemic blood sample;inappropriate blood-to-anticoagulant ratio;estroprogestinic therapy in girls aged 12 and over;BMI >95th percentile for the child’s age and sex or, among older adolescents, a BMI >30.0 kg/m^2^.

### 2.2. Sample Collection and Handling

Venous blood was obtained by peripheral venipuncture, through a 23-G needle, from the antecubital vein, applying only a light tourniquet to avoid stasis. The samples were taken between 7 and 9 a.m. for all patients. Blood was drawn into vacutainer pediatric tubes, which had the same external dimensions as the adult ones, but allowed the sampling of a limited blood volume (1 mL) (Greiner Bio-One, Kremsmünster, Austria) with anticoagulant sodium citrate (0.109 M) at a ratio of 1:9 (*v*/*v*, sodium citrate: blood), according to international recommendations [16]. These tubes have a mark indicating the point to which each tube had to be filled with blood. Gentle, repeated inversion was applied to tubes after blood collection, to ensure perfect anticoagulation.

Blood samples were then handled according to current recommendations for the preanalytical phase [17,18]. Within 4 h of blood collection, platelet poor plasma (PPP) was prepared by centrifugation at 2000× *g*, for 15 min at +20 °C. Double centrifugation significantly reduces the residual amount of platelets and is specifically required for phospholipid-dependent coagulation tests (lupus anticoagulant testing). Aliquots of plasma samples were then frozen and stored at −80 °C until analysis. All blood samples were thawed within 6 months of collection using a 37 °C water bath for a period of time strictly restricted to thawing, and then homogenized by repeated gentle inversion and measured within 2 h.

### 2.3. Laboratory Assays

Samples were analyzed with an STA-R Max^2^ coagulation analyzer (Diagnostica Stago, Asnières sur Seine, France). This system is a fully automated analyzer with mechanical clot detection, designed to run chromogenic and immuno-turbidimetric assays. Parameters measured in the study and related reagents are listed in Table 1.

Prothrombin time (PT) was evaluated using the STA-NeoPTimal reagent prepared from rabbit brain tissue factor with an ISI close to 1 (0.9 to 1.1). PT values are reported as the clotting time in seconds (s). Activated partial thromboplastin time (aPTT) was evaluated using STA-Cephascreen reagent containing cephalin (platelet substitute) and a polyphenolic activator in a buffered medium. aPTT values are reported as the clotting time in seconds (s). Antithrombin activity was measured by a chromogenic assay (STA-STACHROM ATIII). The STA-Stachrom ATIII functional assay is based on the principle of bovine FIIa inhibition in the presence of heparin. AT results are expressed in %. Fibrinogen was determined according to the method of Clauss [19] using the STA-Liquid Fib reagent; this assay is based on the addition of thrombin and calcium to the citrated plasma sample, and the time required to form a clot is then recorded. Fibrinogen results are expressed in mg/dL. The clotting activity of FII, FV, FVII, and FX was measured by PT-based assays, using the respective deficient substrate plasmas. All coagulation factor levels were expressed in %. The clotting activities of FVIII, FIX, FXI, and FXII were measured using aPTT-based assays using the respective deficient substrate plasmas. All coagulation factors levels were expressed in %. VWF antigen was evaluated using the STA-Liatest VWF:Ag reagent. The FXIII assay is an in vitro reagent for the quantitative determination of coagulation in human plasma. Test results are expressed as the percentage of FXIII antigen levels. The FXIII is linear up to 200% with a re-dilution criteria of 1:30 for samples >120%.

D-Dimer (DD) was evaluated using STA-Liatest D-DI Plus reagent. Protein C activity (PC) levels were measured using STA-Stachrom Protein C reagent. Protein S free (PS) antigen levels were measured using an STA-Liatest Free Protein S assay. Resistance to activated Protein C (APC-R) was assayed using a commercial activated partial thromboplastin time-based APC resistance assay, STA-Staclot APC-R). Lupus anticoagulant was detected using the dilute Russell’s viper venom test (STA-DRVV Screen) and a LA sensitive aPTT reagent (STA-PTT-LA). All reagents were from Diagnostica Stago and used within their respective expiration dates. The calibrator used for factor and protein C assays was the STA-Unicalibrator (Diagnostica Stago), which is traceable to the parameter-respective international standard. It contains a known quantity of each factor and is then diluted automatically, to create a calibration curve specific for each assay.

Quality controls were performed using both normal and abnormal control plasmas (STA-System Control N+P, Diagnostica Stago). Bambino Gesù Hospital coagulation laboratory is ISO 9001 certified and ISO 15189 accredited. Analytical procedures did not change over the study period and the laboratory participated in external quality control programs.

### 2.4. Statistical Analysis

Statistical analyses were performed using SPSS statistical software v.17.0 (SPSS Inc., Chicago, IL, USA) and R Language v.3.6.1 (R Foundation for Statistical Computing, Vienna, Austria), with the additional packages dplyr, Hmisc, and boot. Normality distribution was assessed preliminarily by kurtosis, skewness, q-q plot, and Shapiro–Wilk tests. Quantitative variables were expressed by median and interquartile range (IQR), while categorical variables were expressed by absolute and relative frequencies. Data normalization was obtained using Box–Cox transformation with the lambda parameter estimated with the maximum likelihood technique. Outlier detection was performed using Tukey’s method, where an observation was excluded if it was outside the interval Q1 − 1.5*IQR or Q3 + 1.5*IQR, with Q1, Q3, and IQR being respectively the first and the third quartiles and the interquartile range. Reference intervals were obtained using the Harrell–Davis bootstrap method, where the RI was calculated by the Harrell–Davis quantile estimator, a weighted linear combination of order statistics, and a 90% CI by the bootstrap percentile method, using 1000 bootstrap replicates.

## 3. Results

We enrolled a total of 1280 subjects, 671 males (M, 52%) and 609 females (F, 48%), including healthy newborn infants, children and adolescents (0–16 years of age), admitted to our hospital between June 2016 and February 2020. Subjects were subgrouped into five age groups, according to their age: (1) 0–15 days (n = 280 or 22%, 174 M and 106 F), (2) 15–30 days (n = 208 or 16%, 101 M and 107 F), (3) 1–6 months (n = 369 or 29%, 178 M and 191 F), (4) 6–12 months (n = 214 or 17%, 110 M and 104 F), and (5) 1–16 years (n = 209 or 16%, 108 M and 101 F).

The geometric mean of the results of 20 normal samples was used to calculate the ratios for PT, aPTT, PPT-LA, DRVV Screen, and APC-R, as indicated in the CLSI document H47-A2.

The median (IQR) levels of all coagulation factors investigated in the whole sample are shown in Table 2.

The distributions, presence of outliers, and calculation of reference intervals (RI) of the coagulation factors were evaluated in the whole sample (1 group) and after subgrouping subjects, either only for sex (2 subgroups), or only for age classes (5 subgroups), or for all combinations of sex and age (2 × 5 = 10 subgroups). As a result, for each coagulation test, 18 (1 + 2 + 5 + 10) reference intervals were computed and compared (with the exception of the FXIII where only 6 RIs were calculated). To avoid considering valid results of a specific sex or age subgroup as outliers, the presence of possible aberrant values was evaluated within each specific subgroup, before calculating its reference interval.

Figure 1 reports the RIs calculated in all subgroups for PT ratio, aPTT ratio, Fib, TT, and DD.

With few exceptions (F: 0–15 d for aPTT, Fib, DD and TT; M: 6–12 m for TT, DD; M: and F: 1–16 y for TT) the sample size of the 90 subgroups (18 groups × 5 tests) investigated was higher than 50 (Appendix A). For PT and aPTT, the data suggest that from 15 days, a unique RI can be used; whereas in the 0–15 days age class, a different RI is needed (wider RI with a higher upper reference limit) (Figure 1). The PT ratio and aPTT ratio RIs are shown in Figure 1, while the PT and aPTT in seconds are shown in Appendix A. For Fib, a unique RI can be used independently for age and sex (a higher upper reference limit could be considered for the 1–6 months age class). For TT, similar RIs were observed, but RIs were wider and shorter for 0–15 d and 1–16 y classes, respectively. For DD, age specific RIs were observed; indeed, the data show that with increasing age, the size of the RI decreases, without substantial difference between males and females (Figure 1). The differences in RI width were not associated with sample size, since this was similar among the age classes investigated (Appendix A).

Figure 2 reports the RIs calculated in all subgroups for FII, FV, FVII, and FX.

The sample sizes of the subgroups investigated were lower than those observed for the basic coagulation tests, particularly in the age classes <30 days and 1–6 months (Appendix A). Age specific RIs were observed for FII and FX, with increased lower and upper reference limits with increasing age (Figure 2). For FV and FVII similar RIs were noted; however, 0–15 d subgroups showed smaller lower limits (Figure 2). No substantial differences were observed between males and females, with the only exception being the 0–15 d subgroup for FII, FV, and FVII, where for males a shorter RI was evident (Figure 2).

Figure 3 reports the RIs calculated in all subgroups for FVIII, FIX, FXI, FXII, and FXIII.

The sample size of the subgroups investigated were lower than those observed for the basic coagulation tests, particularly in the age class 15–30 days (Appendix A). For FVIII, the RIs were similar, but with wider upper limits for the 0–15 d subgroup. For the FIX, FXI, and FXII age specific RIs were observed, with increased lower and upper reference limits with increasing age (Figure 3). No substantial differences were observed between males and females, with the only exception being the 0–30 d subgroup for FVIII, FIX, FXI, and FXII (Figure 3). For FXIII, data were only available for the class 1–16 y (Appendix A). Due to missing data for the other age classes, the RIs for the specific age class (for males and females) resulted as identical to the RIs calculated in all males and in all females (Appendix A and Figure 3).

The sample sizes for PC and PS were lower than those for AT. Figure 4 reports the RIs calculated in all subgroups for AT, PC, and PS.

Smaller sample sizes (<30) were observed for the 0–15 d and the 6–12 m age classes for PC and PS (Appendix A). Age specific RIs are needed for AT up to 12 months, with increased lower and upper reference limits with increasing age. The RIs for 1–16 years old subjects were shorter, with upper limits similar to those of subjects with 1–12 months. For PC, the data suggest the need for two different types of RIs, one for children up to 6 months (but with a shorter RI for 0–15 days subjects) and a second RI for children between 6 months and 16 years. For PS, the lower and upper limits of the RI increase with age, up to 12 months, but with a shorter RI for the 0–15 day class. No substantial difference was evident between males and females. Interestingly, Figure 4 clearly shows that a common RI was not able to capture the whole variability associated with age.

As expected, the sample sizes for these parameters were lower for all classes, except the classes 1–6 m and 1–16 y (Appendix A). For VWF: Ag and VWF: RCO, the RIs for classes up to 1–6 m were wider and higher, whereas the RIs of subjects of the classes 6–12 months and 1–16 years were shorter and lower (Figure 5).

However, the RIs for the class 15–30 d were not reliable, due to a very small sample size. For PTT-LA, overlapping RIs were observed. For DRVV screening, the RIs of the classes up to 1–6 m were lower than those observed for children >6 months (Figure 5). For APCR, overlapping RIs were observed for classes from 15–30 d to 6–12 m. The RI for class 1–16 y was similar, but with a higher upper limit. Conversely, class 0–15 d displayed a wider and higher RI. The PTT-LA ratio, DRVV Screening ratio, and APCR ratio are shown in Figure 5, while the PTT-LA and DRVV Screening APCR in seconds are shown in Appendix A. For comparison, the RIs for adults in use in our laboratory are reported in Appendix A.

## 4. Discussion

The concept of developmental hemostasis, introduced in 1987 by Andrew et al., is currently widely accepted [20]. The understanding of physiological age-dependent changes in the coagulation system is crucial to achieving an accurate diagnosis in the presence of thrombosis or bleeding disease, especially in newborns or young child. Young children have reduced physiological levels of coagulation proteins, such as factors II, VII, IX, X, XI, and XII and natural coagulation inhibitors, such as antithrombin and protein C and S. However, there are no data supporting an increased risk of thrombosis or bleeding during infancy. Evaluation of pediatric samples requires appropriate reference intervals, in terms of age, analyzer, and reagent [14]. This is particularly necessary when performing tests known to express physiological variances. Laboratories not able to establish their own reference intervals ex novo should use a reference interval established by using the same analyzer and reagent systems (after a validation process). In these cases, population variance should also be taken into account, because population-specific variances [21], reagent-specific variances [22], and analyzer-specific variances [22] have been described. If there is no reference interval value present for the analyzer/reagent combination used in that laboratory, great care should be taken when interpreting coagulation test results in children, and each laboratory should establish its own reference interval for healthy populations in appropriate age groups. When determining age-dependent reference intervals, laboratories should establish groups, in order to standardize age groups [4,5]. In agreement with the CLSI C28-A3, for certain populations, such as newborn, pediatric, and geriatric patients, it may be difficult, if not impossible, to obtain appropriate reference subjects in sufficient numbers. Whatever number of values is obtained, the data should still be analyzed by the nonparametric method and reported as percentiles appropriate to the number of values obtained. The present study supports the concept that adult and child age groups should be evaluated using different reference intervals, to avoid a misdiagnosis, which could lead to serious consequences for the healthcare system, patients, patients’ families, and clinicians [15].

Reference intervals can be estimated using indirect or direct methods. The former use a huge amount of laboratory data and computational algorithms to derive RI, under the hypothesis that the majority of results are from healthy subjects. The latter, instead, calculates the RI using a parametric or non-parametric method on healthy subjects selected by different criteria. Indirect methods, preferred when subjects are difficult to enroll, have been extensively used to estimate pediatric RIs. As an alternative, pediatric RIs have been derived from multicenter studies; however, the heterogeneity among centers (small differences in patient enrolment or in method performances), despite the applications of the same shared study design, could be a major confounding factor in interpreting the results of these studies.

In this study, we used a direct approach by selecting healthy pediatric subjects, using inclusion and exclusion criteria. All age groups included less than 1% of non-Caucasian subjects. A reference interval was calculated in the whole group (1 RI), in the sex-specific groups without taking into consideration age (2 RIs), in the age-specific groups without taking into consideration sex (5 RIs), and in age and sex specific groups (5 × 2 = 10 RIs), for a total of 18 different RIs. All reference intervals were calculated after the exclusion of possible outliers. Since the presence of outliers was verified in each specific group, before RI calculation, it is possible that a result was considered as an outlier in the whole population or when considering only sex, but as a valid observation in a specific age + sex subgroup. Indeed, as shown by the Appendix A, the number of observations considered in the whole population may differ from the sum of observations obtained with the two sex groups or with the five age groups. This choice was made in order to avoid the risk of erroneously labelling valid results of a specific subgroup as outliers, therefore arbitrarily reducing the RI heterogeneity. Given the skewed distribution of data and the moderate to low number of results available for some subgroups, we chose to apply a robust non-parametric approach for RI calculation based on the Harrell–Davis quantile estimator in conjunction with the percentile bootstrap method. With few exceptions, the size of the different subgroups considered was higher than 30, suggesting reliable calculated RIs for many sex and/or age subgroups.

Our results clearly show that among age groups, or in the whole population, few or no substantial differences were observed between males and females. For some parameters, a common RI could be used for all subgroups (Fib, FV, FXIII, PTT-LA ratio), while for others a specific RI for the class 0–15 d should be considered (PT, aPTT, TT, FVII, FVIII). However, for many parameters, age specific RIs are needed (DD, FII, FX, FIX, FXI, FXII, AT, PC, PS, VWF: Ag, VWF: RCO, DRVV Screening ratio, APCR ratio). If observing the RIs obtained for these last parameters, it is clearly evident that the use of a single wider RI for the whole pediatric population (in black color in the figures) could easily lead to misinterpretation of the laboratory results. Our study has some limitations: (1) a reduced sample size for some advanced coagulation tests; (2) RIs were calculated using data obtained in a monocentric study; and (3) RIs were not subjected to an external validation process. The RIs obtained in this study should, therefore, be considered with caution; however, despite the reduced sample size for some groups, this study clearly shows that the definition of age tailored RIs for coagulation tests, at least for some of them, is pivotal and urgent.

Reporting and summarizing such a huge amount of data is difficult and may lead to confusion or misinterpretation; to avoid this, in Table 3, we report a provisional practical proposal for a possible implementation of RIs for the coagulation parameters in the pediatric population.

## Figures and Tables

**Figure 1 diagnostics-12-02552-f001:**
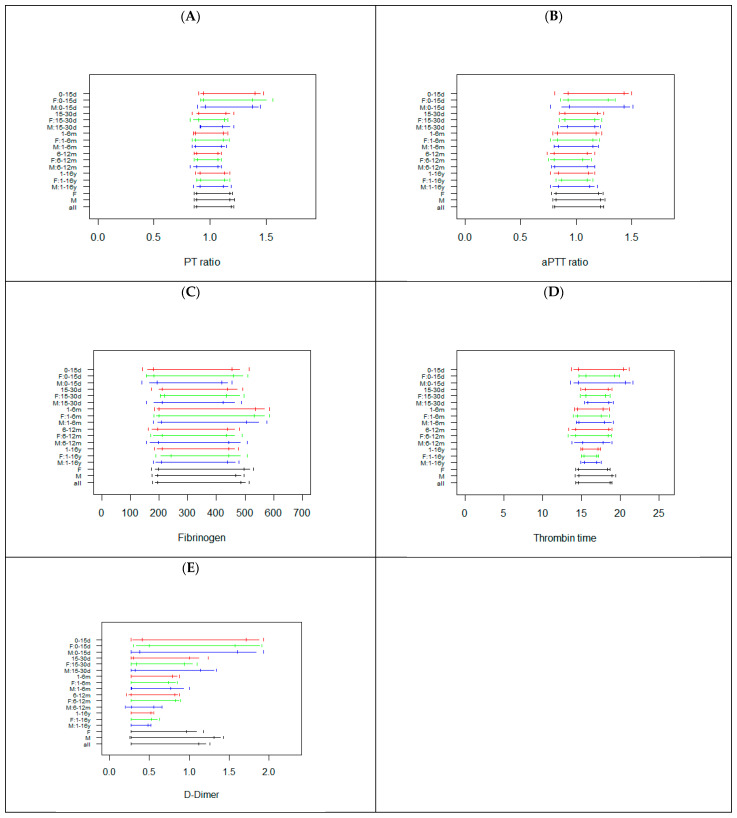
Reference intervals for PT ratio (**A**), aPTT ratio (**B**), Fibrinogen (**C**), TT (**D**), and D–Dimer (**E**) in the subjects subgrouped by sex and age. The horizontal segments represent the RIs, while the vertical segments indicate the limits of the 90% confidence interval of the RIs. For each class, RIs are reported combining males and females (red) or, respectively, only in females (green) or males (blue). The last three RIs are calculated without taking into account age, in females, males, or in the whole sample. d = days, m = months, y = years, F = females, M = males.

**Figure 2 diagnostics-12-02552-f002:**
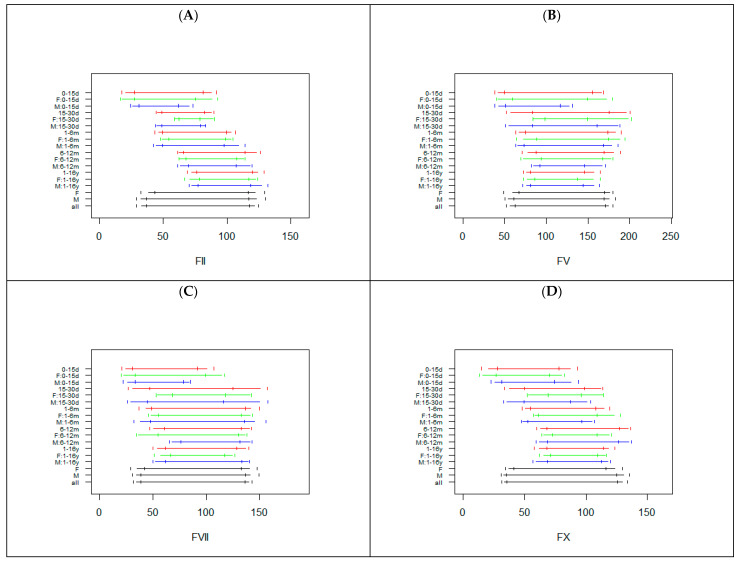
Reference intervals for FII (**A**), FV (**B**), FVII (**C**), and FX (**D**) in the subjects subgrouped by sex and age. The horizontal segments represent the RIs, while the vertical segments indicate the limits of the 90% confidence interval of the RIs. For each class RIs are reported combining males and females (red) or, respectively, only in females (green) or males (blue). The last three RIs are calculated without taking into account age, in females, males, or in the whole sample. d = days, m = months, y = years, F = females, M = males.

**Figure 3 diagnostics-12-02552-f003:**
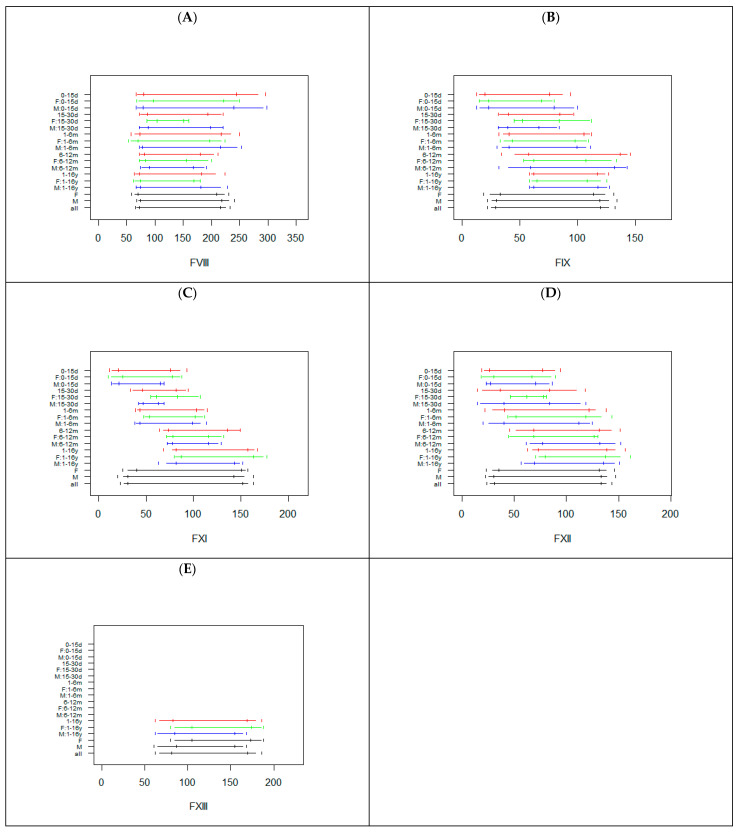
Reference intervals for FVIII (**A**), FIX (**B**), FXI (**C**), FXII (**D**), and FXIII (**E**) in the subjects subgrouped by sex and age. The horizontal segments represent the RIs, while the vertical segments indicate the limits of the 90% confidence interval of the RIs. For each class, RIs are reported combining males and females (red) or, respectively, only in females (green) or males (blue). The last three RIs are calculated, without taking into account age, in females, males, or in the whole sample. d = days, m = months, y = years, F = females, M = males.

**Figure 4 diagnostics-12-02552-f004:**
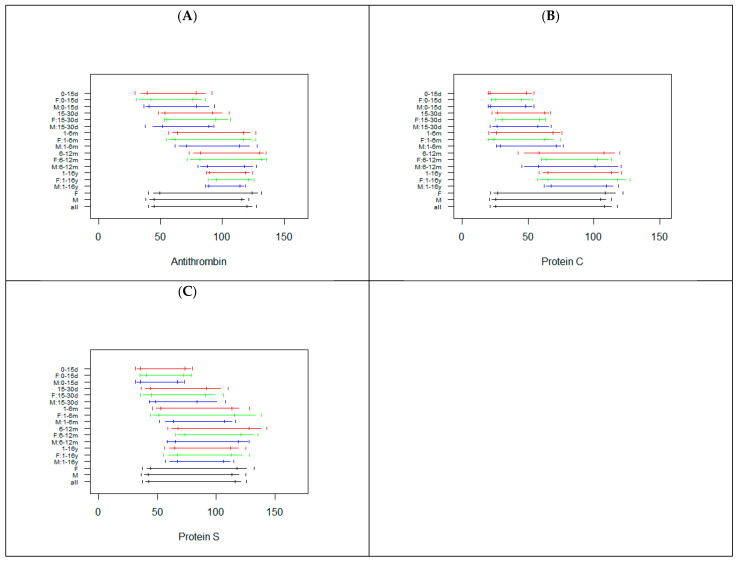
Reference intervals for Antithrombin (**A**), Protein C (**B**), and Protein S (**C**) in the subjects subgrouped by sex and age. The horizontal segments represent the RIs, while the vertical segments indicate the limits of the 90% confidence interval of the RIs. For each class, RIs are reported combining males and females (red) or, respectively, only in females (green) or males (blue). The last three RIs are calculated without taking into account age, in females, males, or in the whole sample. d = days, m = months, y = years, F = females, M = males.

**Figure 5 diagnostics-12-02552-f005:**
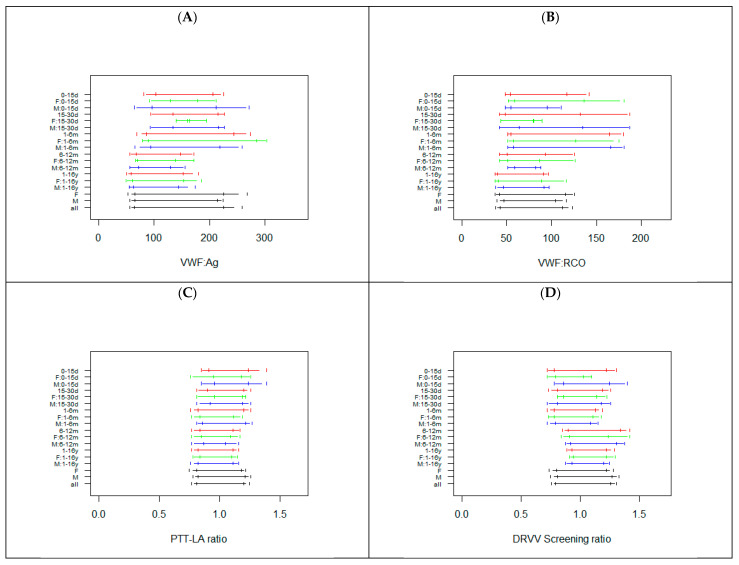
Reference intervals for VWF: Ag (**A**), VWF: RCO (**B**), PTT-LA ratio (**C**), DRVV Screening ratio (**D**), and APCR ratio (**E**) in the subjects subgrouped by sex and age. The horizontal segments represent the RIs, while the vertical segments indicate the limits of the 90% confidence interval of the RIs. For each class, RIs are reported combining males and females (red) or, respectively, only in females (green) or males (blue). The last three RIs were calculated without taking into account age, in females, males, or in the whole sample. d = days, m = months, y = years, F = females, M = males.

**Table 1 diagnostics-12-02552-t001:** List of parameters included in the pediatric reference interval range study, related reagents, type of assay, and reported units. PT: prothrombin time; aPTT: activated partial thromboplastin time; s: seconds.

Parameter	Reagent	Type of Assay	Units
PT	STA-NeoPTimal	Clotting	S
aPTT	STA-CEPHASCREEN	Clotting	S
Fibrinogen (functional)	STA-LIQUID FIB	Clotting	mg/dL
Antithrombin activity	STA-STACHROM AT III	Chromogenic	%
Protein C anticoagulant activity	STA-STACHROM PROTEIN C	Chromogenic	%
Protein S free antigen	STA-LIATEST FREE PROTEIN S	Immuno-turbidimetric	%
Thrombin time	STA-THROMBIN	Clotting	S
D-Dimer	STA-LIATEST D-DI PLUS	Immuno-turbidimetric	µg/mL
Von Willebrand Factor antigen	STA-LIATEST VWF:AG	Immuno-turbidimetric	%
Von Willebrand Factor RCO	VWF:RCO	Immuno-turbidimetric	%
Lupus anticoagulant (screening)	PTT-LA and STA-STACLOT DRVV SCREEN	Clotting	Ratio
Extrinsic pathway factors	STA-DEFICIENT II, V, VII and X	Clotting	%
Intrinsic pathway factors	STA-IMMUNODEF VIII, IX, XI and XII	Clotting	%
Factor XIII	K-Assay Factor XIII	Immuno-turbidimetric	%
Detection of activated Protein C resistance	STACLOT APC-R	Clotting	S

**Table 2 diagnostics-12-02552-t002:** Levels of coagulation factors, expressed as the median and interquartile range (IQR) in the whole sample investigated.

Factor (Unit)	Descriptive Statistics	Factor (Unit)	Descriptive Statistics
PT (s) PT ratio	13.5 (12.9–14.6) 0.99 (0.95–1.07)	Factor VIII (%)	120 (99–159)
aPTT (s) aPTT ratio	32.8 (30.3–35.2) 1.01 (0.93–1.08)	Factor IX (%)	71 (53–90)
Fibrinogen (mg/dL)	298 (250–367)	Factor X (%)	79 (64–95)
D-Dimer (µg/mL)	0.51 (0.35–0.74)	Factor XI (%)	85 (58–106)
Anti-thrombin (%)	89 (68–103)	Factor XII (%)	83 (59–110)
Thrombin time (s)	16.5 (15.9–17.3)	Factor XIII (%)	121 (110–144)
Protein C (%)	55 (38–81)	VWF Ag (%)	131 (95–167)
Protein S (%)	79 (64–93)	VWF RCO (%)	74 (61–90)
Factor II (%)	78 (62–95)	APCR (s) APCR ratio	147 (130–165) 1.00 (0.88–1.12)
Factor V (%)	110 (93–133)	PTT LA (s) PTT LA ratio	40.6 (37.5–44.2) 1.01 (0.93–1.10)
Factor VII (%)	87 (68–107)	DRVV (s) DRVV ratio	32.0 (29.5–35.0) 1.00 (0.92–1.09)

**Table 3 diagnostics-12-02552-t003:** Practical proposal to implement age and/or sex specific RIs for coagulations tests.

Test	Proposal
PT, aPTT	Use specific RI for the group 0–15 d
Fib, TT	Use a common RI (for TT use wider RI for the group 0–15 d)
DD	Use age specific RIs
FII, FX	Use age specific RIs
FV, FVII	Use a common RI (for FVII use shorter RI for the group 0–15 d)
FVIII	Use a common RI (wider RI for the group 0–15 d)
FIX, FXI, FXII	Use age specific RIs
FXIII	Use a common RI (data only available for the class 1–16 y)
AT, PC, PS	Use age specific RIs
VWF: Ag, VWF: RCO	Use age specific RIs
PTT-LA ratio	Use a common RI
DRVV Screening ratio	Use a common RI for age up to 6 m and a different common RI for age >6 m
APCR ratio	Use age specific RIs

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
