# Peer review of "Reference Intervals for Coagulation Parameters in Developmental Hemostasis from Infancy to Adolescence"

_diagnostics, 2022, doi:10.3390/diagnostics12102552_

Round 1
Reviewer 1 Report
This study addresses a well needed objective, i.e. reference interval for coagulation assays in children. However, I have a few queries
Did the authors look for intraindividual and interindividual differences?
Did the authors consider the circadian variation?
Did authors consider different weight of the same age group children?
Author Response
Dear Reviewer
attached you can find the final version of our manuscript with all the changes made.
Best regards
Giovina Di Felice

Reviewer 2 Report
The work from di Felice et al, is an important and elegant addition to knowledge on reference intervals in hemostasis laboratories.
However, several concerns should be addressed:
· Several coagulation factor levels are affected by ethnicity. Even if a sub-analysis of this age dependent RI, is not possible (or necessary) it would be appreciated if a description is presented. This is especial in line with the authors own limitation Nr3 on external validation.
· With respect to the assays used, it would be important to state that the FXIII assay measures the antigen and not the activity as the other factor assays do. This could be misleading when readers are not familiar with the K-Assay.
· The authors measured FXIII values up to 185. However, the publicly available package insert states an assay range up to 140%. It would be good for the authors to state the procedure implemented in case of values above the measuring range.
· The authors report ratios for PT, aPTT, PPT-LA, DRVV Screen and APC. The authors should state the method or reference value they used to generate these ratios.
· The authors always refer to use a common RI for infancy and adolescence. Most laboratories only have implemented a refence interval for the adult, wither by the manufactures or ideally established locally. It would be an important addition to provide the RI for adults in the test system used by the authors. This could improve the understanding and translation into other laboratory settings and thus improve young patient mamagement.
· The authors should harmonize the nomenclature of factors. FII vs. Factor FXII (figure 3), VWF:RCO vs RCO (figure 5)
In addition, there some Italian lines in the text, which need to be deleted as they are already translated (e.g. line 183-184, line 300-301)
Author Response
Dear Reviewer,
attached you can find the final version of our manuscript with all the changes made.
Best regards
Giovina Di Felice

Round 2
Reviewer 2 Report
Thank you very much for the additions made.
I am looking forward seeing the article published and referenced.